# Nobiletin Prevents D-Galactose-Induced C2C12 Cell Aging by Improving Mitochondrial Function

**DOI:** 10.3390/ijms231911963

**Published:** 2022-10-08

**Authors:** Hui-Hui Wang, Ya-Nan Sun, Tai-Qi Qu, Xue-Qin Sang, Li-Mian Zhou, Yi-Xuan Li, Fa-Zheng Ren

**Affiliations:** 1College of Food Science and Engineering, Gansu Agricultural University, Lanzhou 730070, China; 2Beijing Advanced Innovation Center for Food Nutrition and Human Health, Department of Nutrition and Health, China Agricultural University, Beijing 100193, China

**Keywords:** nobiletin, C2C12 myoblast, mitochondrial function, ROS, aging

## Abstract

Age-associated loss of skeletal muscle mass and function is one of the main causes of the loss of independence and physical incapacitation in the geriatric population. This study used the D-galactose-induced C2C12 myoblast aging model to explore whether nobiletin (Nob) could delay skeletal muscle aging and determine the associated mechanism. The results showed that Nob intervention improved mitochondrial function, increased ATP production, reduced reactive oxygen species (ROS) production, inhibited inflammation, and prevented apoptosis as well as aging. In addition, Nob improved autophagy function, removed misfolded proteins and damaged organelles, cleared ROS, reduced mitochondrial damage, and improved skeletal muscle atrophy. Moreover, our results illustrated that Nob can not only enhance mitochondrial function, but can also enhance autophagy function and the protein synthesis pathway to inhibit skeletal muscle atrophy. Therefore, Nob may be a potential candidate for the prevention and treatment of age-related muscle decline.

## 1. Introduction

Skeletal muscle is the largest tissue, accounting for nearly 40% of the total body mass in humans, and is the most important motor organ [1,2]. Meanwhile, skeletal muscle is also the largest mitochondria-rich metabolic organ with crucial roles in activity, thermogenesis, and overall energy homeostasis [3]. With an increase in age, the loss of skeletal muscle mass and function predisposes older adults to substantial health risks, including insulin resistance and diabetes, and results in reduced quality of life and an increased incidence rate and mortality in the elderly [4,5,6].

Aging is an unavoidable process that gradually leads to functional decline in almost all organisms [7]. Cellular senescence is a hallmark and the main driver of aging [8,9,10]. Over time, the accumulation of senescent cells contributes to an increasing list of pathologies in tissues and accelerates the aging of tissues and organs [7]. There are many causes inducing cellular senescence, including oxidative stress, DNA damage, telomere shortening [11,12], and inflammation [13]. Homeostasis of senescent cells changes markedly, disrupting mitochondrial electron transport chain, resulting in the leakage of ROS [10]. Excessive ROS can impair mitochondria and nuclear DNA, resulting in a positive feedback loop to reinforce cell cycle arrest [10]. In addition, ROS also accelerate the rate of telomere shortening [14], promote inflammation and apoptosis [13], and accelerate aging.

Nobiletin (Nob) is a polymethoxyl flavonoid found in some citrus peels and has been reported to have multiple functions [15], including antioxidant [16] and anti-inflammatory functions [15,17], improvement of mitochondrial function, and maintenance of metabolic homeostasis [3]. A previous study has reported that Nob enhanced antioxidant activity to clear ROS and MDA production and alleviate nonalcoholic fatty liver disease in rats fed with a high-fat diet [16]. Nohara et al. found that Nob could reduce ROS and improve and optimize mitochondrial function in skeletal muscle [3]. D-galactose-induced (D-gal-induced) oxidative stress is a good model for studying aging [18,19]. Hence, we hypothesized that Nob intervention would improve skeletal muscle atrophy during the aging process. Based on previous findings, our study established the C2C12 myoblast aging model by D-gal to explore whether Nob could delay, alleviate, or prevent the aging process and its underlying mechanism. This study provides direct evidence for the beneficial role of Nob in the prevention and treatment of skeletal muscle dysfunction.

## 2. Results

### 2.1. Effect of Nobiletin (Nob) on D-Galactose-Induced (D-Gal-Induced) Atrophy of Skeletal Muscle

In this study, we used D-gal-induced C2C12 cells to mimic the atrophy of skeletal muscle during the aging process [20,21]. The treatment concentration of D-gal was chosen at 20 mg/mL according to Chen’s research [21]. The optimal concentration and time of Nob treatment were determined by cell viability and the key markers of senescence (Appendix A). To better characterize the effect of Nob on the atrophy of skeletal muscle, we performed MHC immunofluorescence staining of myotubes. The myotube area in the D-gal-induced group was significantly smaller than the control (CK) group. In contrast, Nob intervention significantly improved myotube area (by about 20%) compared with the D-gal group (Figure 1A,B). The initiation of terminal differentiation and fusion of myoblasts begins with the expression of myogenin [22]. In this study, D-gal induction significantly decreased the expression of myogenin compared with the CK group, while Nob intervention promoted increased expression of myogenin and the differentiation and fusion of myoblasts (Figure 1C). The adjustment of muscle fiber size, due to its limited proliferative capacity, is determined by the coordinated balance between protein synthesis and protein degradation [6]. In order to further understand the underlying mechanisms of Nob in skeletal muscle atrophy, we examined the signaling pathway involved in protein synthesis (Akt/mTOR). Therefore, we evaluated the expression of p-S473-Akt/Akt and p-p70 S6K/p70 S6K. We found that D-gal induction significantly reduced the ratios of p-S473-Akt/Akt and p-p70 S6K/p70 S6K compared to the CK group (*p* < 0.05). However, compared with the D-gal group, Nob treatment significantly restored the ratios of p-S473-Akt/Akt and p-p70 S6K/p70 S6K, which increased by 56.2% and 22.6%, respectively (Figure 1C). Myofibrillar proteins account for approximately 55–60% of total muscle proteins, and the ubiquitin–proteasome system degrades mostly myofibrillar proteins, so we also examined the expression of ubiquitin protein [6,23]. Our studies indicated that ubiquitin was markedly increased in the D-gal-induced aging model relative to the CK group. The elevated ubiquitin level was significantly attenuated by Nob treatment (Figure 1D). Moreover, we also found that the expression of MAFbx and MuRF1 were significantly increased in the D-gal-induced group. However, Nob treatment attenuated down-regulated expression of MAFbx and MuRF induced by D-gal (Figure 1D). Taken together, Nob improved D-gal-induced muscle fiber atrophy by balancing protein synthesis with protein degradation.

### 2.2. Effect of Nob on Senescent Cells and Senescence Markers in D-Gal-Induced C2C12 Cells

To examine whether Nob could alleviate, delay, or prevent D-gal-induced C2C12 myoblast aging, we measured SA-β-gal staining in D-gal-induced C2C12 cells treated with DMSO or Nob for 48 h. We observed a significant increase in SA-β-gal^+^ cells after D-gal treatment relative to the CK group (*p* < 0.05) and a distinct decrease in the number of SA-β-gal^+^ cells in the Nob intervention group (*p* < 0.05) (Figure 2A,B). Next, we examined several senescence-associated markers at both the protein and mRNA levels. Compared with the CK group, D-gal induction led to a dramatic increase in the expression of P16 protein (*p* < 0.05), whereas Nob strongly reversed the D-gal-induced expression of P16 protein (Figure 2C,D,F). Furthermore, consistent with the P16 protein levels, P53 and P21 exhibited similar results to P16 at both the protein and mRNA expression levels (Figure 2E,F).

### 2.3. Effect of Nob on ROS and Inflammation in D-Gal-Induced C2C12 Cells

In addition to ATP, mitochondrial respiration also generates ROS [3,10]. If not controlled, increased ROS cause cellular damage and accelerate senescence [10,12]. Therefore, we investigated whether Nob could reduce the level of ROS and inflammation in the D-gal-induced aging model. We found that D-gal induction significantly increased ROS levels (Figure 3A–C), down-regulated the master antioxidant regulator nuclear factor erythroid 2-related factor 2 (NRF2) (Figure 3D), and increased expression of p-P65/P65 (inflammation) relative to the CK group (Figure 3D). In contrast, Nob intervention markedly reduced the level of ROS, enhanced the antioxidant capacity, and partly recovered the level of inflammation to CK levels (Figure 3D).

### 2.4. Effect of Nob on Mitochondrial Function in D-Gal-Induced C2C12 Cells

Mitochondrial function is an important factor determining skeletal muscle function [24]. Mitochondria are considered the powerhouse of the cells and they undertake key energy generators [1]. Moreover, mitochondria are crucial for regulating the cellular metabolism, cell cycle, and apoptosis [10,25]. Meanwhile, mitochondria serve as the main producer of ROS. Increased ROS contribute to impaired mitochondrial function, ultimately leading to senescence during aging [10,26]. Therefore, we next examined the effect of Nob on mitochondrial function under the D-gal-induced oxidative stress aging model. ATP production in the D-gal-induced aging model group was significantly decreased relative to the CK group (about 40%), yet partly recovered by Nob treatment (*p* < 0.05) (Figure 4C). Likewise, whereas D-gal-induced treatment impaired basal respiration, maximum respiration, and spare respiratory capacity of mitochondrial respiration compared to the CK group, Nob partly reversed these parameters to normal levels (Figure 4A,B,D,E). The decrease in MMP is a landmark event in the early stage of apoptosis [27]. Next, we measured fluorescence intensity of JC-1 monomer by flow cytometry. Our findings demonstrated that D-gal induction significantly increased JC-1 monomer and the ratio of Bax/Bcl-2 (*p* < 0.05), indicating activation of mitochondrial-mediated apoptosis (Figure 4F,G). Nob treatment prevented D-gal-induced apoptosis, which partly recovered to the CK group’s levels. Together, these results indicate a robust efficacy of Nob to improve mitochondrial function and prevent apoptosis.

### 2.5. Effect of Nob on Autophagy in D-Gal-Induced C2C12 Cells

In order to understand the underlying mechanisms of Nob for the prevention of D-gal-induced C2C12 cell aging, we examined the signaling pathway involved in autophagy-dependent catabolism. AMPK-activated kinase (AMPK) has a vital role in the regulation of autophagy [28,29]. Therefore, we evaluated the expression of phosphorylated AMPK at Thr172 (p-AMPK) and AMPK in CK and D-gal-induced C2C12 myoblasts treated with DMSO or Nob. We found that D-gal induction markedly reduced the expression of p-T172-AMPK/AMPK protein compared with the CK group; Nob treatment partly restored the expression of p-T172-AMPK/AMPK proteins to normal levels (Figure 5). We also examined autophagy-associated protein expression of LC3 II/I and LAMP2. LC3 I is the cytosolic form, which is further converted to an autophagosome-associated form, LC3 II [29]. Therefore, the LC3 II/I ratio is used as an indicator to measure autophagic activity. Consistent with p-T172-AMPK/AMPK proteins, D-gal-induced expression of LC3 II/I proteins revealed an obvious decrease relative to the CK group. However, Nob treatment significantly increased the expression of LC 3II/I proteins (Figure 5). Furthermore, consistent with LC3 II/I protein levels, LAMP2 showed similar results to LC3 II/I expression levels (Figure 5). These results suggested that Nob may activate the autophagy signaling pathway to prevent aging.

## 3. Discussion

Aging is an age-related progressive deterioration course with reduced physical and functional potential due to oxidative stress, chromatin modifications, or oncogene activation, amongst other forms of stress, which leads to an increased risk of diseases and ultimately results in death [7,9]. Sarcopenia is a geriatric syndrome characterized by the aging-related loss of muscle mass and function and can significantly increase the risk of poor outcomes [21]. However, there are currently no effective interventions to counteract age-associated muscle decline [30,31]. With the increasing aging of the global population, it is important to find some safe and effective food-derived substances to intervene and treat sarcopenia. This study explored the fact that Nob, a natural flavonoid, can improve the area of myotubes by regulating protein synthesis and degradation. The new study builds on findings from previously reported research on Nob intervention in older mice that demonstrated the functional impact of Nob on mitochondrial health in a D-galactose-induced (D-gal-induced) C2C12 myoblast aging model [3]. The present study also found that Nob can activate AMPK-mediated autophagy, clear mitochondrial damage, and improve skeletal muscle atrophy. Therefore, these results provide strong evidence that Nob prevents and treats age-related muscle decline.

The D-gal-accelerated oxidative stress aging model has been widely accepted [18]. It is based on metabolic theory and exhibits performance similar to that of the natural aging process in many aspects [18]. It is well-known that D-gal can induce oxidative stress and lead to skeletal muscle atrophy [32,33]. Kou et al. found that the injection of D-gal at a dose of 150 mg/kg/day in Sprague–Dawley rats for 8 weeks could significantly reduce the gastrocnemius muscle weight/body weight ratio and cross-sectional area of muscle fibers compared with the CK group [29]. SA-β-gal is the first and still the most widely used senescent marker [34]. Senescent cells often show increased lysosomal β-gal activity and secrete pro-inflammatory cytokines with potent effects [9]. In addition, the most prominent characteristic of senescence is durable growth arrest. Therefore, senescent cells show an up-regulation of genes that enforce cell cycle arrest, such as P16, P21, and P53 [10,35]. Chen et al. treated differentiated C2C12 cells with D-gal (0, 10, 20, and 40 mg/mL) for 48 h. They found that SA-β-gal^+^ cells and the expressions of P16 and P53 proteins significantly increased after 20 mg/mL D-gal-induced treatment [21]. Consistent with previous studies, our studies observed that the number of SA-β-gal^+^ cells and the expression of senescence-associated proteins (P16, P21, and P53) were significantly increased in the D-gal-induced aging model when compared with the CK group (Figure 2). However, there was a significant decrease in the number of SA-β-gal^+^ cells and suppression of senescence-associated protein (P16, P21, and P53) expression in the presence of Nob (Figure 2). Thus, our data suggest that D-gal could induce C2C12 aging and Nob intervention could delay, alleviate, or prevent D-gal-induced aging.

Skeletal muscle is mitochondria-rich due to its extraordinary demand for ATP. Mitochondria generate ATP through oxidative phosphorylation of carbohydrates and fatty acids [36]. ROS are by-products of mitochondrial oxidative phosphorylation, generated mainly by electron leakage in the ETC [10]. Mitochondrial DNA plays an important role in maintaining and regulating mitochondrial function, but because its physical location is close to the mitochondrial electron transport, it is very vulnerable to ROS [37]. Previous studies have found that mitochondria from senescent cells exhibit decreased MMP, increased proton leakage, and ROS [26,36]. Another study demonstrated that inhibiting ROS could prevent senescence in aged satellite cells [38]. It is worth noting that mitochondrial homeostasis is vital for maintaining the viability and function of C2C12 myoblasts. In the current study, D-gal induction significantly increased ROS levels, while Nob intervention significantly improved mitochondrial function and reduced the generation of ROS (Figure 3 and Figure 4). Nob exerted profound efficacies to promote D-gal-induced C2C12 cell mitochondrial function as evidenced by partially reversing basal respiration, ATP production, maximum respiration, and spare respiratory capacity to normal levels (Figure 5A–E). The decrease in MMP is a landmark event in the early stage of apoptosis [27]. In our study, D-gal significantly increased JC-1 monomer and the expression of Bax/Bcl-2 (*p* < 0.05), while Nob prevented D-gal-induced apoptosis, and partly recovered to CK levels (Figure 4F,G).

We also found that Nob treatment improved autophagy function in the D-gal-induced aging model. Autophagy plays a crucial role in maintaining skeletal muscle homeostasis [17]. Previous research found that autophagy-deficient mice exhibited skeletal muscle atrophy, increased dysfunctional mitochondria, and excessive apoptosis [39,40]. Autophagy loss leads to dysfunctional organelles, misfolded proteins, and the accumulation of ROS. It is well-known that LC 3 is vital for the formation of early autophagic vacuole membranes [17]. Our research showed that Nob intervention significantly increased the expression of LC 3II/I and LAMP2 protein under D-gal-induced C2C12 cells (Figure 5). These results suggested that Nob may activate the autophagy signaling pathway to remove misfolded proteins and damaged organelles and, thus, reduce intracellular ROS. Therefore, appropriate autophagy function is essential for improving mitochondrial function through the clearance of ROS and dysfunctional or damaged mitochondria [17,41]. Furthermore, Nob promoted D-gal-induced C2C12 cell health aging as evidenced by enhancing the antioxidant and anti-inflammatory capacity. Nob can also improve the area of myotubes by regulating protein synthesis and degradation. These results indicated a robust efficacy of Nob to enhance health to counteract skeletal muscle aging.

## 4. Materials and Methods

### 4.1. Materials

Dulbecco’s modified Eagle medium (DMEM, C11995500BT), fetal bovine serum (10099141C), and horse serum (160501222) were purchased from Thermo Fisher Scientific (Waltham, MA, USA). Nobiletin (Nob) (IN0210) was purchased from Solarbio Technology Company (Beijing, China). D-galactose (D-gal) (ST1219) was purchased from Beyotime Biotechnology Company (Shanghai, China).

### 4.2. Cell Culture and Treatment

C2C12 myoblasts were purchased from the Institute of Basic Medicine, Chinese Academy of Medical Sciences. C2C12 myoblasts were incubated in DMEM medium containing 10% fetal bovine serum (FBS) and 1% penicillin/streptomycin (P/S) in a humidified incubator with 5% CO_2_ at 37 °C. The C2C12 myoblast aging model was established via D-gal (20 mg/mL) treatment for 48 h. Nob (10 μM) was dissolved in DMSO and added to the medium together with D-gal. When the cells had grown to 50–60% confluency, they were treated as follows: (1) Control (CK) group; (2) D-gal group; (3) D-gal + Nob group.

To obtain differentiated myotubes, C2C12 myoblasts were incubated in DMEM medium with 10% FBS and 1% P/S until 80–90% confluency. Then, C2C12 myoblasts were maintained in differentiation medium (DM) for differentiation, and then treated as follows: (1) CK group; (2) D-gal group; (3) D-gal + Nob group. Nob (10 μM) was dissolved in DMSO and added to the DM together with D-gal (20 mg/mL). DM was changed every two days until the cells were fully differentiated (around day 5).

### 4.3. Cell Viability Assay

C2C12 myoblasts were seeded in 96-well plates at a density of 5 × 10^3^ cells/well until they reached 50–60% confluency. The CK and D-gal-induced C2C12 myoblasts were then treated for 48 h with DMSO or Nob. Cell viability was measured using the cell counting kit-8 (CK04) according to the manufacturer’s instructions (Solarbio, Beijing, China). The absorbance of each well was measured at 450 nm with a microplate reader.

### 4.4. Senescence-Associated β-Galactosidase (SA-β-Gal) Activity

SA-β-gal staining was detected using the senescence β-galactosidase staining kit (C0605) according to the manufacturer’s protocols (Beyotime, Shanghai, China). SA-β-gal^+^ C2C12 cells were quantified as a percentage of the total number of cells.

### 4.5. Immunofluorescence Staining

The culture medium was removed, and the cells were washed twice with phosphate-buffered saline (PBS). The cells were then fixed at room temperature with 4% paraformaldehyde/PBS for 30 min. They were washed 3 × 5 min and then treated with 0.2% Triton X-100/PBS for 10 min. After that, they were washed 3 × 5 min and then blocked in 10% sheep serum/PBS at room temperature for 60 min. Slides were then incubated in 10% sheep serum/PBS with primary antibody overnight at 4 °C. They were washed 3 × 5 min, and then incubated with the secondary antibody (1:500) in the dark at room temperature for 1 h. The slides were sealed with a DAPI-containing anti-fluorescence quencher (Solarbio, Beijing, China) and imaged with laser confocal microscopy (Leica).

### 4.6. RT-qPCR: RNA Extraction, cDNA Synthesis, and PCR

Total RNA was extracted using the TRIzol method (Invitrogen). Extracted RNA was used for cDNA synthesis according to the manufacturer’s instructions (abm, Canada) of the cDNA synthesis kit (G592). The cDNA was used for RT-qPCR with PowerUp™ SYBR™ Green (A25742, Thermo Fisher Scientific, Waltham, MA, USA). The relative expression of target genes was normalized by the housekeeping gene GAPDH. The following primers were used: *P53*, forward: GATGACTGCCATGGAGGAGT, reverse: CTCGGGTGGCTCATAAGGTA; *P21*, forward: CCAGGCCAAGATGGTGTCTT, reverse: TGAGAAAGGATCAGCCATTGC; GAPDH, forward: TGGCCTTCCGTGTTCCTAC reverse: GAGTTGCTGTTGAAGTCGCA.

### 4.7. Western Blotting

The culture medium was removed, and after washing the cells twice with PBS, they were harvested for further analysis. The preparation of cell lysates and Western blotting were performed by the method of Sun et al. [42]. Antibodies used were p16 INK4A (F-12) (Santa Cruz Biotechnology, sc-1661), P53 mouse monoclonal antibody (Proteintech, 60283-2-Ig), P21 rabbit polyclonal antibody (Proteintech, 27296-1-AP), LC3B antibody (Cell Signaling, 2575S), LAMP2 (Abcam, ab13524), AMPK (Cell Signaling, D63G4), P-AMPK (Cell Signaling, 2535S), p70 S6 Kinase Antibody (Cell Signaling, 9202S), p-p70 S6 Kinase Antibody (Cell Signaling, 9234S), and GAPDH polyclonal antibody (Proteintech, 10494-1-AP).

### 4.8. ROS Measurements

The treated cells were incubated with a 10 uM DCFH-DA fluorescent probe (CA1410) for 20 min at 37 °C, following the manufacturer’s protocols (Solarbio, Beijing, China) and directly analyzed with or without fixing by laser confocal microscopy (Leica) or flow cytometry (BD). For mean fluorescence intensity (MFI) determination, we used ImageJ or the flow cytometry analysis software Flowjo. MFI refers to the total fluorescence intensity divided by the total area or the number of total cells.

### 4.9. Seahorse Mitochondrial Oxygen Consumption Rate Measurements

After CK and D-gal-induced C2C12 myoblast cells had been treated for 48 h with DMSO or Nob, they were measured for mitochondrial oxygen consumption rate (OCR) by the Seahorse XF24 Extracellular Flux Analyzer (Seahorse Bioscience, North Billerica, MA, USA) according to the manufacturer’s instructions for the XF Cell Mito Stress Test Kit (No.103150-100, Seahorse Bioscience). The rate measurement equations are as follows: basal respiration = (last rate measurement before first injection)—(non-mitochondrial respiration rate); ATP production = (last rate measurement before oligomycin injection)—(minimum rate measurement after oligomycin injection); maximum respiration = (maximum rate measurement after FCCP injection)—(non-mitochondrial respiration rate); spare respiration capacity = (maximum respiration)—(basal respiration).

### 4.10. JC-1 Measurements

The mitochondrial membrane potential (MMP) was measured using flow cytometry. CK and D-gal-induced C2C12 myoblast cells were treated for 48 h with DMSO or Nob. They were then used for JC-1 (C2003S) fluorescent probe staining according to the manufacturer’s protocols (Beyotime, Shanghai, China). Fluorescence was determined at 514/529 nm (excitation/emission, green) and 585/590 (excitation/emission, red) using flow cytometry (BD).

### 4.11. Statistical Analysis

All statistical analyses were performed using Excel 2020 to calculate each index’s mean value and standard error. All data were analyzed using one-way ANOVA followed by Duncan’ post-hoc test. *p* < 0.05 was considered to be statistically significant. All figures were plotted with GraphPad Prism 8. ROS and P16 images were taken by laser confocal microscopy (Leica).

## 5. Conclusions

In conclusion, D-gal induced C2C12 myoblast aging causes skeletal muscle atrophy. Nob improved mitochondrial function, reduced ROS production, enhanced antioxidant and anti-inflammatory capacity, and prevented apoptosis. Nob-activated autophagy played an obvious protective role in D-gal-induced atrophy of skeletal muscle. Together, our study illustrates that Nob can improve mitochondrial function, prevent aging, and promote healthy aging.

## Figures and Tables

**Figure 1 ijms-23-11963-f001:**
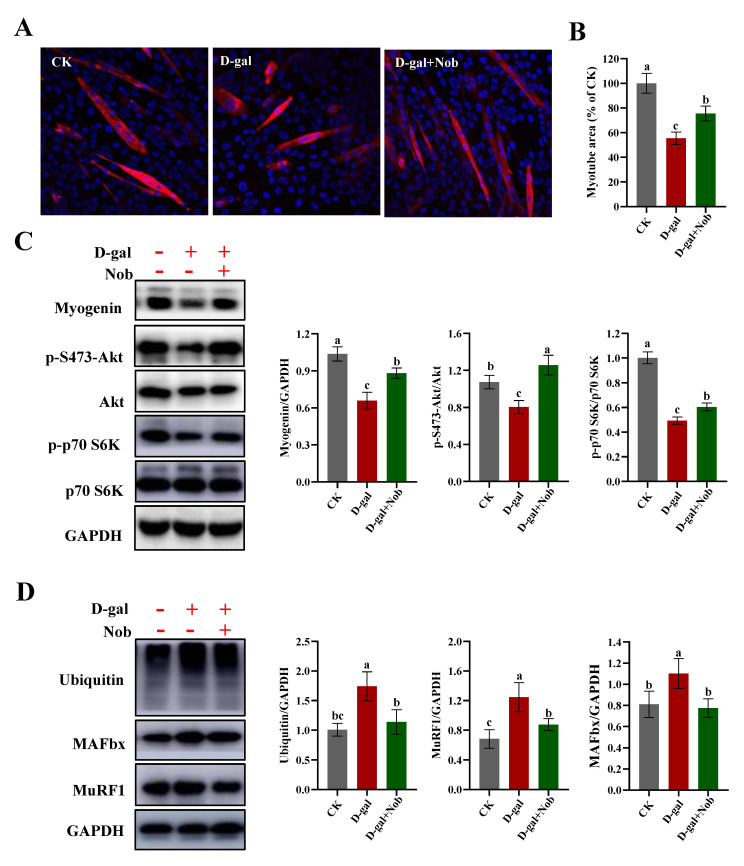
Nob improved D-gal-induced atrophy of skeletal muscle. (**A**) Immunostaining of myotube myosin heavy chain (MHC) in CK and D-gal-induced differentiated C2C12 cells treated for 5 days with DMSO or Nob. (**B**) Quantification of the myotube area in CK and D-gal-induced C2C12 myoblasts treated for 5 days with DMSO or Nob. (**C**,**D**) Western blot analysis of myogenin, p-S473-Akt, Akt, p-p70 S6K, p70 S6K, ubiquitin, MAFbx, MuRF1, and GAPDH in CK and D-gal-induced differentiated C2C12 cells treated for 5 days with DMSO or Nob. Different lowercase letters represent significant differences between different treatment groups (*p* < 0.05).

**Figure 2 ijms-23-11963-f002:**
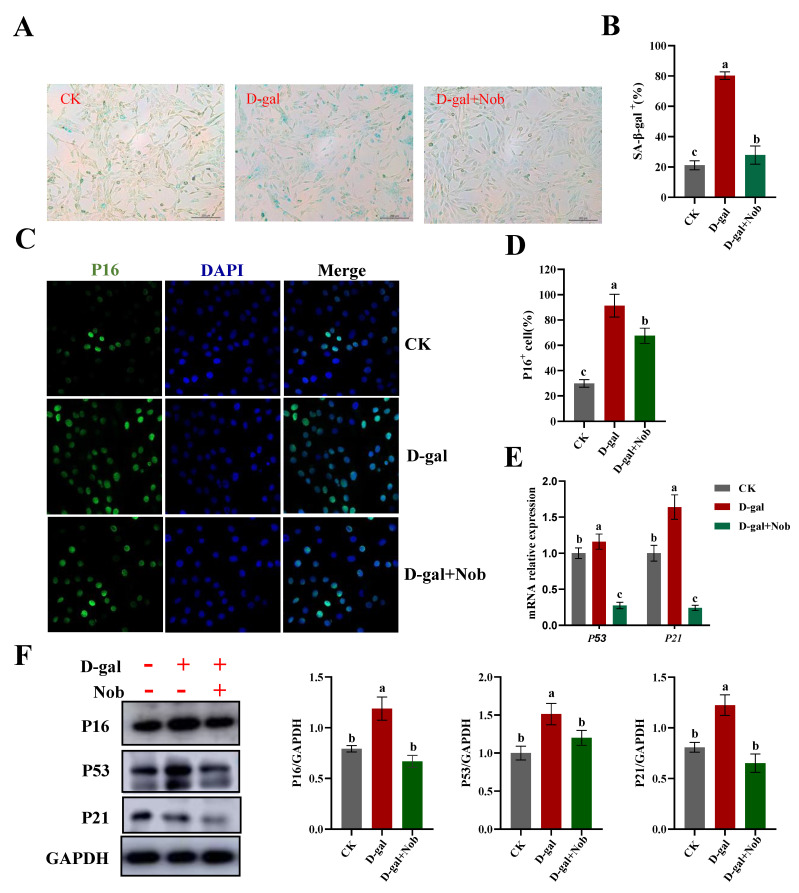
Nob decreased the expression of senescence markers in D-gal-induced senescent cells. (**A**,**B**) Representative images of SA-β-gal staining in CK and D-gal-induced C2C12 myoblasts treated for 48 h with DMSO or Nob. Scale bar, 200 μm. (**C**,**D**) Immunostaining of P16 in CK and D-gal-induced C2C12 myoblasts treated for 48 h with DMSO or Nob. Representative images are shown. Scale bar, 25 μm. (**E**) mRNA of P53 and P21 in CK and D-gal-induced C2C12 myoblasts treated for 48 h with DMSO or Nob. (**F**) Western blot analysis of P16, P53, P21, and GAPDH in CK and D-gal-induced C2C12 myoblasts treated for 48 h with DMSO or Nob. Different lowercase letters represent significant differences between different treatment groups (*p* < 0.05).

**Figure 3 ijms-23-11963-f003:**
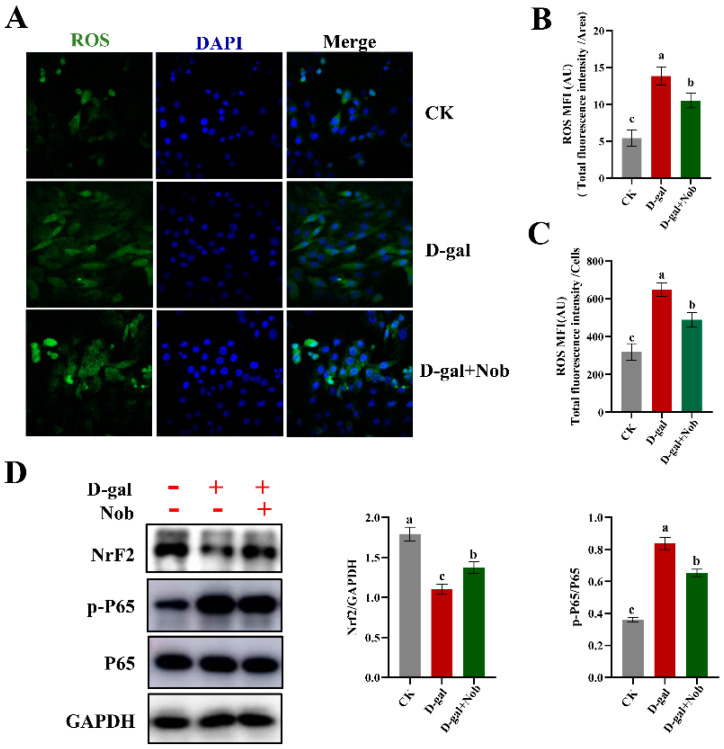
Nob decreased levels of ROS and inflammation in D-gal-induced senescent cells. (**A**,**B**) Quantification of ROS levels by laser confocal microscopy in CK and D-gal-induced C2C12 myoblasts treated for 48 h with DMSO or Nob. Representative images are shown. Scale bar, 25 μm. (**C**) Quantification of ROS levels by flow cytometry in CK and D-gal-induced C2C12 myoblasts treated for 48 h with DMSO or Nob. (**D**) Western blot analysis of Nrf2, p-P65, P65, and GAPDH in CK and D-gal-induced C2C12 myoblasts treated for 48 h with DMSO or Nob. Different lowercase letters represent significant differences between different treatment groups (*p* < 0.05).

**Figure 4 ijms-23-11963-f004:**
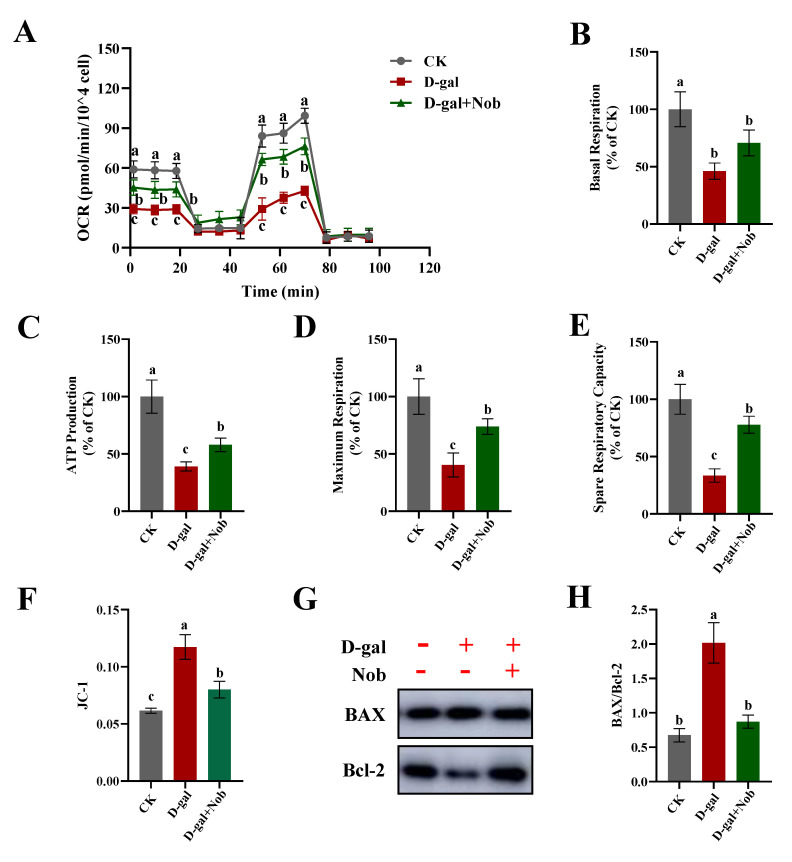
Nob improved mitochondrial function and prevented apoptosis in D-gal-induced senescent cells. (**A**) Oxygen consumption rate (OCR), (**B**) basal respiration, (**C**) ATP production, (**D**) maximum respiration, and (**E**) spare respiratory capacity were measured by the XF Cell Mito Stress Test Kit in CK and D-gal-induced C2C12 myoblasts treated with DMSO or Nob. (**F**) Analysis of fluorescence intensity of JC-1 monomer (green) by flow cytometry in CK and D-gal-induced C2C12 myoblasts treated with DMSO or Nob. (**G**,**H**) Western blot analysis of BAX and Bcl-2 in CK and D-gal-induced C2C12 myoblasts treated with DMSO or Nob. Different lowercase letters represent significant differences between different treatment groups (*p* < 0.05).

**Figure 5 ijms-23-11963-f005:**
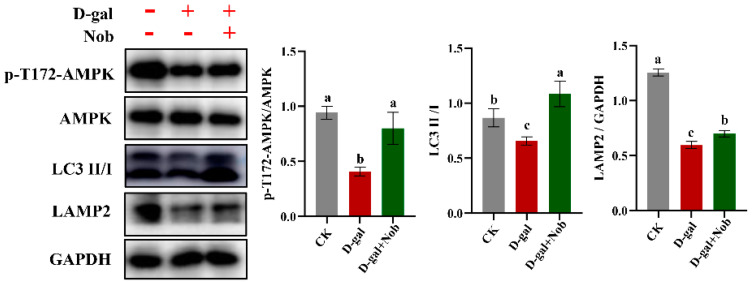
Nob improved autophagy in D-gal-induced senescent cells. Western blot analysis of p-T172-AMPK, AMPK, LC3 II/I, LAMP2, and GAPDH in CK and D-gal-induced C2C12 myoblasts treated with DMSO or Nob. Different lowercase letters represent significant differences between different treatment groups (*p* < 0.05).

## Data Availability

The data presented in this study are available in the article and Appendix A.

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
