# Peer review of "Nobiletin Prevents D-Galactose-Induced C2C12 Cell Aging by Improving Mitochondrial Function"

_ijms, 2022, doi:10.3390/ijms231911963_

Round 1

Reviewer 1 Report

The work of Hui-Hui Wang and colleagues delves into the positive effects on mitochondrial function of the flavonoid nobiletin. The article focuses on the protective effects of loss of muscle energy (and trophic) function. The paper is comprehensive and well described; the groups and controls of the experiments are well designed, although this paper mainly focus on the protecting effect of nobiletin on muscle aging, but not on its possible preventive effect on muscle aging, a topic that could be played out by a series of future experiments and  also mentioned in the discussion. The use of the word "prevention" in the title of the article is only related to the D-galactose-induced cell aging (an in vitro cell aging model).

Just a note about the letters a,b,c placed on the columns of the figure graphs (significance of the data? p value?) I could not find any references.

Author Response

Dear editor and the reviewer,

Thank you very much for the positive evaluation of our manuscript entitled “Nobiletin prevents D-galactose-induced C2C12 cell aging by improving mitochondrial function” (ijms-1917351). We appreciate the constructive and helpful comments of reviewers; they helped to greatly improve our manuscript. To address all points raised by the reviewers, we added new content in revised manuscript. The texts of our responses are marked in blue font in this file. Revised text was also shown in blue font in the revised manuscript file, and the exact line numbers where changes were pointed out in the responses.

We hope that our revised manuscript is now suitable for publication in IJMS.

Thank you for the consideration.

Sincerely,

Fazheng Ren

The responses to reviewer’s comments and the main corrections in the paper are as follows:

Reviewer reports:

Reviewer #1:

The work of Hui-Hui Wang and colleagues delves into the positive effects on mitochondrial function of the flavonoid nobiletin. The article focuses on the protective effects of loss of muscle energy (and trophic) function. The paper is comprehensive and well described; the groups and controls of the experiments are well designed, although this paper mainly focus on the protecting effect of nobiletin on muscle aging, but not on its possible preventive effect on muscle aging, a topic that could be played out by a series of future experiments and also mentioned in the discussion. The use of the word "prevention" in the title of the article is only related to the D-galactose-induced cell aging (an in vitro cell aging model).

Just a note about the letters a, b, c placed on the columns of the figure graphs (significance of the data? p value?) I could not find any references.

--Response: Thank you for carefully reading our manuscript and giving us valuable suggestion, we feel very lucky and grateful for your approval about our research. We quite agree to your comments and suggestions about the manuscript. We feel sorry for the neglect about the details in our manuscript. We had added P value and explanation of a, b, and c in all figure legends.

Reviewer 2 Report

Comments to the authors:

1.  It is not enough to observe the nobiletin effect only based on one single mouse skeletal muscle cell type, C2C12 cells; at least, primary human skeletal muscle cells should be applied as well.

2. The whole results are based on in vitro experiment. In vivo animal model is highly suggested since anti-ageing models are commonly used in the related studies if the authors aimed to present nobiletin as a new compound in the prevention or treatment of ageing.

3. The results are mostly phenotypic, observative and descriptive. Some mechanistic studies should be performed.

Minor

1. The novelty and innovative potential of your manuscript compared to the published literature should be described in more detail in the abstract and discussion section.

2.  A positive control has to be included as a reference standard

3. In Figure 1C and D, label the western blot analyses properly. If possible, replace the Akt and p70 S6K blots with suitable ones.  

4.  In line 42: Expand Nob

5.  In line 58: “process[21,22]” needs to correct as “process [21,22]”.

6.  In line 64: Expand CK group

7.  In line 72: “[6][6].” needs to correct as “[6]”.

8.  In line 72: “autophagy[29,30].” needs to correct as “autophagy [29,30]”.

9.  In line 269: “5×103 cells/well” needs to correct as “5×103 cells/well”.

Author Response

Dear editor and the reviewers,

Thank you very much for the positive evaluation of our manuscript entitled “Nobiletin prevents D-galactose-induced C2C12 cell aging by improving mitochondrial function” (ijms-1917351). We appreciate the constructive and helpful comments of reviewers; they helped to greatly improve our manuscript. To address all points raised by the reviewers, we added new content in revised manuscript. The texts of our responses are marked in blue font in this file. Revised text was also shown in blue font in the revised manuscript file, and the exact line numbers where changes were pointed out in the responses.

We hope that our revised manuscript is now suitable for publication in IJMS.

Thank you for the consideration.

Sincerely,

Fazheng Ren

The responses to reviewer’s comments and the main corrections in the paper are as follows:

  1. It is not enough to observe the nobiletin effect only based on one single mouse skeletal muscle cell type, C2C12 cells; at least, primary human skeletal muscle cells should be applied as well.

--Response: Thank you for carefully reading our manuscript and giving us constructive and helpful comment and suggestion. We quite agree to your comment and suggestion about the manuscript. Due to ethical issues, it is very difficult for us to obtain human muscle samples. But thank you very much for your suggestion, and we will consider doing it in the future. In addition, C2C12 cell line is very representative in skeletal muscle research. This cell line can proliferate in vitro and differentiate under specific conditions. It had been widely used in skeletal muscle related research, such as: [1] Nohara, K.; Mallampalli, V.; Nemkov, T.; Wirianto, M.; Yang, J.; Ye, Y.Q.; Sun, Y.X.; Han, L.; Esser, K.A.; Mileykovskaya, E.; et al. Nobiletin fortifies mitochondrial respiration in skeletal muscle to promote healthy aging against metabolic challenge. Nat Commun 2019, 10. [2] Chen, Q.N.; Fan, Z.; Lyu, A.K.; Wu, J.; Guo, A.; Yang, Y.F.; Chen, J.L.; Xiao, Q. Effect of sarcolipin-mediated cell transdifferentiation in sarcopenia-associated skeletal muscle fibrosis. Exp Cell Res 2020, 389, 111890. [3] Sun, Y.N.; Yang, Z.X.; Ren, F.Z.; Fang, B. FGF19 alleviates palmitate-induced atrophy in C2C12 cells by inhibiting mitochondrial overload and insulin resistance. Int J Biol Macromol 2020, 158, 401-407.

  1. The whole results are based on in vitro experiment. In vivo animal model is highly suggested since anti-ageing models are commonly used in the related studies if the authors aimed to present nobiletin as a new compound in the prevention or treatment of ageing.

--Response: We do appreciate you for giving us constructive comment and pointing out this critical issue. In vivo animal experiments have also been considered before, and this part of experiments is under way, and these results will be published soon. Thank you again.

  1. The results are mostly phenotypic, observative and descriptive. Some mechanistic studies should be performed.

--Response: Thanks for your constructive comment. We are sorry that we did not describe it clearly. Here are our experimental ideas: Oxidative stress is one of the main causes of skeletal muscle aging, and D-galactose-induced aging model is also due to the increase of oxidative stress in the body. First of all, we found that Nob can alleviate C2C12 aging (such as, SA-β-gal staining, P16, P53, P21, etc.). Therefore, we speculated that the mechanism of its alleviation aging is to reduce reactive oxygen species (ROS) production and increase the removal of ROS. Because mitochondria serve as the main producer of ROS. We explored the role of Nob in enhancing mitochondrial function and found that it can increase the capacity of mitochondrial electron transport chain (seahorse). In additional, since autophagy is the primary way to clear ROS, we detected autophagic flow in cells and found that Nob can improve autophagy and accelerate ROS clearance. Together, these mechanisms coordinately prevent the aging of D-galactose-induced C2C12 cells, namely reduce ROS production by improving mitochondrial function and accelerate ROS clearance by increasing
autophagy.

Article idea map:

Minor

  1. The novelty and innovative potential of your manuscript compared to the published literature should be described in more detail in the abstract and discussion section.

--Response: Thanks for your constructive comment. We quite agree to your comment and suggestion about the manuscript. Our results illustrated that Nob can not only enhance mitochondrial function, but also enhance autophagy function and protein synthesis pathway to inhibit skeletal muscle atrophy. Therefore, these results provide strong evidence for Nob to prevent and treat age-related muscle decline. We had added more details in the abstract and discussion section.

The added text as follows:

In line 20-23 (Abstract): Moreover, our results illustrated that Nob can not only enhance mitochondrial function, but also enhance autophagy function and protein synthesis pathway to inhibit skeletal muscle atrophy. Therefore, Nob may be a potential candidate for the prevention and treatment of age-related muscle decline.

In line 205-217 (Discussion): Sarcopenia is a geriatric syndrome characterized by the aging-related loss of muscle mass and function and can significantly increase the risk of poor outcomes [22]. However, there are currently no effective interventions to counteract age-associated muscle decline [31,32]. With the increasing aging of the global population, it is important to find some safe and effective food-derived substances to intervene and treat sarcopenia. This study explored that Nob, a natural flavonoid, can improve the area of myotubes by regulating protein synthesis and degradation. The new study builds on findings from previously reported research with Nob intervention older mice that demonstrated both the functional impact of Nob on mitochondrial health in D-galactose-induced (D-gal-induced) C2C12 myoblast aging model [3]. The present study also found that Nob can activate AMPK mediated autophagy, clear mitochondrial damage and improved the of skeletal muscle atrophy. Therefore, these results provide strong evidence for Nob to prevent and treat age-related muscle decline.

[3] Nohara, K.; Mallampalli, V.; Nemkov, T.; Wirianto, M.; Yang, J.; Ye, Y.Q.; Sun, Y.X.; Han, L.; Esser, K.A.; Mileykovskaya, E.; et al. Nobiletin fortifies mitochondrial respiration in skeletal muscle to promote healthy aging against metabolic challenge. Nat Commun 2019, 10.

[22] Chen, Q.N.; Fan, Z.; Lyu, A.K.; Wu, J.; Guo, A.; Yang, Y.F.; Chen, J.L.; Xiao, Q. Effect of sarcolipin-mediated cell transdifferentiation in sarcopenia-associated skeletal muscle fibrosis. Exp Cell Res 2020, 389, 111890.

[32] Kwak, J.Y.; Kwon, K.-S. Pharmacological Interventions for Treatment of Sarcopenia: Current Status of Drug Development for Sarcopenia. Ann Geriatr Med Res 2019, 23, 98-104.

[33] Pal, R.; Palmieri, M.; Loehr, J.A.; Li, S.; Abo-Zahrah, R.; Monroe, T.O.; Thakur, P.B.; Sardiello, M.; Rodney, G.G. Src-dependent impairment of autophagy by oxidative stress in a mouse model of Duchenne muscular dystrophy. Nat Commun 2014, 5, 4425.

  1. A positive control has to be included as a reference standard.

--Response: We do appreciate you for giving us constructive comment and pointing out this critical issue. There are currently no effective interventions to counteract age-associated muscle decline [1-3]. Some studies had reported that increasing protein intake seemed to be the most effective intervention to prevent the loss of skeletal muscle mass and physical function. However, there were still some studies had found that increasing protein intake cannot improve physical function or muscle mass in the geriatric population [4,5]. At present, the most effective intervention is resistance training, but its application is still blocked in the geriatric population [5]. Therefore, there is no properly positive control.

[1]. Denison, H.J., Cooper, C., Sayer, A.A., Aihie Sayer, A., and Robinson, S.M. (2015). Prevention and optimal management of sarcopenia: a review of combined exercise and nutrition interventions to improve muscle outcomes in older people. Clin. Interventions Aging 10, 859–869.

[2]. Kwak, J.Y., and Kwon, K.-S. (2019). Pharmacological interventions for treatment of sarcopenia: current status of drug development for sarcopenia. Ann. Geriatr. Med. Res. 23, 98–104.

[3]. Singh, A.; D'Amico, D.; Andreux, P.A.; Fouassier, A.M.; Blanco-Bose, W.; Evans, M.; Aebischer, P.; Auwerx, J.; Rinsch, C. Urolithin A improves muscle strength, exercise performance, and biomarkers of mitochondrial health in a randomized trial in middle-aged adults. Cell Rep Med 2022, 3.

[4]. Björkman, M.P .; Suominen, M.H.; Kautiainen, H.; Jyväkorpi, S.K.; Finne-Soveri, H.U.; Strandberg, T.E.; Pitkälä, K.H.; Tilvis, R.S. Effect of Protein Supplementation on Physical Performance in Older People With Sarcopenia–A Randomized Controlled Trial. J. Am. Med. Dir. Assoc. 2020, 21, 226–232.

[5]. Ganapathy, A.; Nieves, J.W. Nutrition and Sarcopenia—What Do We Know? Nutrients 2020, 12, 1755.

  1. In Figure 1C and D, label the western blot analyses properly. If possible, replace the Akt and p70 S6K blots with suitable ones.

--Response: Thanks for your constructive comment. The ratio of p-S473-Akt/Akt and p-p70 S6K/p70 S6K reflect protein synthesis signal pathway activation [1, 2]. When the expression of Akt and p70 S6K remains unchanged, the phosphorylation degree represents the activation degree of protein synthesis signal pathway. In our study, D-gal induction decreased the ratio of p-S473-Akt/Akt and p-p70 S6K/p70 S60K compared with the control group, indicating that its protein synthesis was inhibited. In contrast, Nobiletin intervention markedly increased the ratio of p-S473-Akt/Akt and p-p70 S6K/p70 S60K relative to D-gal group, indicating Nob could improve D-gal-induced protein synthesis.

[1] Kou, X.; Li, J.; Liu, X.; Yang, X.; Fan, J.; Chen, N. Ampelopsin attenuates the atrophy of skeletal muscle from D-gal-induced aging rats through activating AMPK/SIRT1/PGC-1α signaling cascade. Biomedicine & Pharmacotherapy 2017, 90, 311-320.

[2] Caldow, M.K.; Ham, D.J.; Trieu, J.; Chung, J.D.; Lynch, G.S.; Koopman, R. Glycine Protects Muscle Cells From Wasting in vitro via mTORC1 Signaling. Front Nutr 2019, 6.

  1. In line 42: Expand Nob

--Response: We are truly grateful to your helpful comment and suggestion. We had revised the “Nob” into “Nobiletin (Nob)” in line 44.

  1. In line 58: “process[21,22]” needs to correct as “process [21,22]”.

--Response: We do thank you for your kindly suggestion. We had revised the “process[21,22]” into “process [21,22]” in line 61.

  1. In line 64: Expand CK group

--Response: We do thank you for your helpful suggestion. We feel sorry for the neglect about the format details in our manuscript. We had revised the “CK group” into “control (CK) group” in line 67.

  1. In line 72: “[6][6].” needs to correct as “[6]”.

--Response: Thank you for carefully reading our manuscript and giving us helpful comment. We had revised the “[6][6]” into “[6]” in line 74.

  1. In line 172: “autophagy[29,30].” needs to correct as “autophagy [29,30]”.

--Response: We do thank you for your kindly suggestion. We must apologize for me giving an imperfect manuscript. We had revised the “autophagy[29,30].” into “autophagy [29,30]” in line 178.

  1. In line 269: “5×103 cells/well” needs to correct as “5×103 cells/well”.

--Response: We feel sorry for the neglect about the format details in our manuscript and we sincerely thank the reviewer for the comments and corrections. We have revised the “5×103 cells/well” into “5×103 cells/well” in line 296.

Reviewer 3 Report

It was shown that Nobiletin (Nob) could reduce ROS and improve and optimize mitochondrial function in skeletal muscle (3). D-galactose-induced (D-gal-induced) oxidative stress is a good model for studying aging (19, 20). In this manuscript, the authors explore the effect of Nob on the skeletal muscle atrophy during the aging process by D-gal in differentiated C2C12 myoblasts.

They showed that

1)   Nob ameliorated D-gal-induced atrophy of skeletal muscle (immunostaining of myotube myosin heavy chain) accompanied with the balancing between protein synthesis (Akt/mTOR-signal) and protein degradation (ubiquitin and atrogene level) (figure 1).

2)   Nob could alleviate SA-β-gal staining senescence-associated markers expression in D-gal induced C2C12 cells (figure 2).

3)   Nob reduced the level of DCFH-DA fluorescent (ROS) and expression of P-P65/P65 (inflammation) in D-gal induced C2C12 cells (figure 3).

4)   Nob improved mitochondrial function and prevented apoptosis in D-gal-induced senescent cells (figure 4).

5)   Nob improved autophagy in D-gal-induced senescent cells (figure5).

Based on the above results they propose that Nob can optimize mitochondrial function and prevent skeletal muscle aging. Although the exact mechanism how oxidative stress affect autophagy, this work is a nice contribution to the field of the role of Nob in D-galactose-induced C2C12 cell.

I have only few concerns to address as below.

Major points

1.    Are all the effects of Nob on D-galactose-induced C2C12 cell due to its anti-oxidant? I understand Nob could reduce ROS and improve and optimize mitochondrial function in skeletal muscle, leading improvement of myotube area. What is the exact mechanism of Nob on the autophagy function?

2.    P < 0.05 was considered to be statistically significant in the Materials and Methods section. However, no explanation of a, b, and c in all figure legends.

Author Response

Dear editor and the reviewers,

Thank you very much for the positive evaluation of our manuscript entitled “Nobiletin prevents D-galactose-induced C2C12 cell aging by improving mitochondrial function” (ijms-1917351). We appreciate the constructive and helpful comments of reviewers; they helped to greatly improve our manuscript. To address all points raised by the reviewers, we added new content in revised manuscript. The texts of our responses are marked in blue font in this file. Revised text was also shown in blue font in the revised manuscript file, and the exact line numbers where changes were pointed out in the responses.

We hope that our revised manuscript is now suitable for publication in IJMS.

Thank you for the consideration.

Sincerely,

Fazheng Ren

The responses to reviewer’s comments and the main corrections in the paper are as follows:

Reviewer:

It was shown that Nobiletin (Nob) could reduce ROS and improve and optimize mitochondrial function in skeletal muscle (3). D-galactose-induced (D-gal-induced) oxidative stress is a good model for studying aging (19, 20). In this manuscript, the authors explore the effect of Nob on the skeletal muscle atrophy during the aging process by D-gal in differentiated C2C12 myoblasts.

They showed that

1) Nob ameliorated D-gal-induced atrophy of skeletal muscle (immunostaining of myotube myosin heavy chain) accompanied with the balancing between protein synthesis (Akt/mTOR-signal) and protein degradation (ubiquitin and atrogene level) (figure 1).

2) Nob could alleviate SA-β-gal staining senescence-associated markers expression in D-gal induced C2C12 cells (figure 2).

3) Nob reduced the level of DCFH-DA fluorescent (ROS) and expression of P-P65/P65 (inflammation) in D-gal induced C2C12 cells (figure 3).

4) Nob improved mitochondrial function and prevented apoptosis in D-gal-induced senescent cells (figure 4).

5) Nob improved autophagy in D-gal-induced senescent cells (figure5).

Based on the above results they propose that Nob can optimize mitochondrial function and prevent skeletal muscle aging. Although the exact mechanism how oxidative stress affect autophagy, this work is a nice contribution to the field of the role of Nob in D-galactose-induced C2C12 cell.

I have only few concerns to address as below.

Major points

  1. Are all the effects of Nob on D-galactose-induced C2C12 cell due to its antioxidant? I understand Nob could reduce ROS and improve and optimize mitochondrial function in skeletal muscle, leading improvement of myotube area. What is the exact mechanism of Nob on the autophagy function?

--Response: Thanks for your constructive comment. In our study, D-galactose-induced oxidative stress could lead to C2C12 cells aging, and Nobiletin inhibits D-galactose-induced C2C12 cells aging via its antioxidant function. In this study, Nobiletin mainly acts through antioxidant, but Nobiletin also has anti-inflammatory function. In Figure 3D, we had detected the classic inflammatory signal pathway (NF-kB). The results showed that Nobiletin intervention significantly reduced the expression of p-P65/P65 compared to D-gal group. On the one hand, Nobiletin inhibited protein degradation of C2C12 cells through NF-kB /MURF1 or MAFbx pathway. On the other hand, Nobiletin also reduced the production of senescence-associated secretory phenotype (SASP) (IL-6, TNF-α) through inhibiting NF-kB signal pathway [1].

Autophagy is an evolutionary conserved process to maintain cellular homeostasis. Its role is to deliver damaged organelles and misfolded proteins to lysosomes for degradation, thus preventing waste accumulation. AMPK/mTORC1/ULK1 is the main major regulatory pathways. Nob significantly increased the ratio of p-T172-AMPK/AMPK compared to D-gal group (Figure. 5). This result indicated that Nob could activate AMPK/mTORC1/ULK1 signaling pathway. Activation of AMPK can promote autophagy through inhibiting mTORC1 and then activating autophagy initiation proteins ULK1 and AGT13 [2].

[1] Lipina, C.; Hundal, H.S. Lipid modulation of skeletal muscle mass and function. J Cachexia Sarcopeni 2017, 8, 190-201.

[2] Parzych, K.R.; Klionsky, D.J. An Overview of Autophagy: Morphology, Mechanism, and Regulation. Antioxid Redox Sign 2014, 20, 460-473.

  1. P < 0.05 was considered to be statistically significant in the Materials and Methods section. However, no explanation of a, b, and c in all figure legends.

--Response: We feel sorry for the neglect about the details in our manuscript and we sincerely thank the reviewer for the comments and corrections. We had added P value and explanation of a, b, and c in all figure legends.

Reviewer 4 Report

In this study the authors attempt to assess the role of nobiletin in the prevention of D-gal- induced senescence in C2C12 myoblasts.

1. The manuscript contains some grammar and syntax errors that need to be corrected (for examples, please refer to the uploaded file).

2. All figures need to be replaced by figures of higher resolution.

3. “Merage” needs to be replaced by “merge”.

4. Figure 4A lacks statistical analysis.

5. Abbreviations should be defined the first time they appear and then be used throughout the text (for examples, please refer to the uploaded file).

6. Given that senescent cells are characterized by their resistance to apoptosis, how is it explained by the authors that both phenomena are induced by D-gal in their model?

7. The major problem in my opinion is that the followed protocol used to induce senescence in C2C12 myoblasts after D-gal treatment is not properly described. Do the authors only treat cells with D-gal once for 48 h and then assess for the markers of senescence? That would mean that the observed phenomena could be rather an immediate and probably transient response of the cells to the treatment than an established senescent phenotype. In addition, when nobiletin is added? Are the cells pre-treated with nobiletin before being treated with D-gal? Are nobiletin and D-gal applied simultaneously? If D-gal treatment is short-term, this could give an explanation for the extremely low percentages of SA-beta Gal- and p16-positive cells presented in Fig. 2B and 2C. This should be checked by the authors, since a 0.8% percentage of SA-beta Gal-positive cells and a 1% percentage of p16-positive cells cannot a support the manifestation of senescence and the existence of an aging model.

Author Response

Dear editor and the reviewers,

Thank you very much for the positive evaluation of our manuscript entitled “Nobiletin prevents D-galactose-induced C2C12 cell aging by improving mitochondrial function” (ijms-1917351). We appreciate the constructive and helpful comments of reviewers; they helped to greatly improve our manuscript. To address all points raised by the reviewers, we added new content in revised manuscript. The texts of our responses are marked in blue font in this file. Revised text was also shown in blue font in the revised manuscript file, and the exact line numbers where changes were pointed out in the responses.

We hope that our revised manuscript is now suitable for publication in IJMS.

Thank you for the consideration.

Sincerely,

Fazheng Ren

The responses to reviewer’s comments and the main corrections in the paper are as follows:

Reviewer:

In this study the authors attempt to assess the role of nobiletin in the prevention of D-gal- induced senescence in C2C12 myoblasts.

  1. The manuscript contains some grammar and syntax errors that need to be corrected (for examples, please refer to the uploaded file).

--Response: We do thank you for your kindly suggestion. We had revised some grammar and syntax errors you pointed out. In addition, we also had checked the grammar and logic of the full text, and revised text was shown in blue font in the revised manuscript file.

In line 18: We had revised “inhibitd” to “inhibited”.

In line 18: We had revised “prevent” to “prevented”.

In line 35-36: We had revised “leads to more pathology” to “contribute to an increasing list of pathologies”.

In line 39: We had revised “mitochondrial electron transport chain (ETC)” to “mitochondrial electron transport chain”.

In line 60: We had revised “skeletal muscle” to “the skeletal muscle”.

In line 72: We had revised “skeletal muscle” to “the skeletal muscle”.

In line 74: We had revised the “[6][6]” into “[6]”.

In line 67-68: We had revised “In contrast, the increased myotube area observed in the presence of Nob pretreatment revealed about 20% increase in Nob treated group compared with the D-gal-induced aging group (Figure. 1A and 1B)” to “In contrast, Nob intervention significantly improved myotube area (about 20%) compared with the D-gal group (Figure. 1A and 1B)”.

In line 77-81: We had revised “We found that Nob treatment significantly improved the expression of p-S473-Akt/Akt and p-p70 S6K/p70 76 S6K when compared with the D-gal-induced group (Figure 1C).” to “We found that D-gal induction significantly reduced the ratio of p-S473-Akt/Akt and p-p70 S6K/p70 S6K compared to the CK group (P<0.05). However, compared with D-gal group, Nob treatment significantly improved the ratio of p-S473-Akt/Akt and p-p70 S6K/p70 S6K, which increased by 56.2% and 22.6% respectively (Figure 1C).” in line 79-83.

In line 117: We had revised “P53 and P21” to “P53 and P21”.

In line 129: We had revised “P-P65/P65” to “p-P65/P65”.

In line 178: We had revised the “autophagy[29,30].” into “autophagy [29,30]”.

In line 244: We had revise “ETC” to “mitochondrial electron transport” in line 244.

In line 278: We had revise Nobiletin to “Nobiletin (Nob)” in line 278.

  1. All figures need to be replaced by figures of higher resolution.

--Response: We do appreciate you for giving us helpful suggestion. We had revised all the figures to high resolution figures.

  1. “Merage” needs to be replaced by “merge”.

--Response: We do thank you for your kindly suggestion. Meanwhile, we must apologize for my carelessness and giving spelling mistakes. We had revised “Merage” to “Merge” in Figure 2C and Figure 3A.

  1. Figure 4A lacks statistical analysis.

-Response: We do thank you for your kindly suggestion. We had added statistical analysis in Figure 4A. The revised figure is as follows:

  1. Abbreviations should be defined the first time they appear and then be used throughout the text (for examples, please refer to the uploaded file).

--Response: Thank you for carefully reading our manuscript and giving us valuable comment and suggestion. We had defined some abbreviations when they appear the first time, and revised text was also shown in blue font in the revised manuscript file. In addition, we also had checked the abbreviations of full text.

In line 44: We had revised “Nob” to “Nobiletin (Nob)”.

In line 64: We had revised “CK group” to “control (CK) group”.

In line 278: We had revised “Nobiletin” to “Nobiletin (Nob)”.

  1. Given that senescent cells are characterized by their resistance to apoptosis, how is it explained by the authors that both phenomena are induced by D-gal in their model?

--Response: Thank you for carefully reading our manuscript and giving us constructive suggestion, we quite agree to your comment about the manuscript. We did find that some senescent cells type had the capacity of senescent cells to resist apoptosis through literature review [1]. For example, Marcotte et al. found that senescent fibroblasts were resistant to apoptotic death [2], and Hampel et al. also reported that senescent fibroblasts were highly resistant to apoptosis [3]. However, different cell types may have different relationships with aging and apoptosis. One of the reasons of aging skeletal muscle mass loss is an increase in apoptosis [4]. Consistent with our result, Kou et al. found that the ratio of Bax/Bcl-2 was significantly increased in D-gal-induced aging rats [5]. In additional, Chen et al also found that the number of apoptotic cells was notably increased in D-gal-induced aging C2C12 myotubes [6]. Siu et al. suggested that Bax/Bcl-2 ratio was significantly increased after H2O2-induced C2C12 myotubes 48h [7]. Whitman et al. had found a significant increase of apoptotic myonuclei in the vastus lateralis muscle of old subjects compared to young controls [8].

[1]. Jeyapalan, J.C.; Sedivy, J.M. Cellular senescence and organismal aging. Mech Ageing Dev 2008, 129, 467-474.

[2] Marcotte, R., Lacelle, C., Wang, E., 2004. Senescent fibroblasts resist apoptosis by downregulating caspase-3. Mech. Ageing Dev. 125, 777–783.

[3] Hampel, B., Wagner, M., Teis, D., Zwerschke, W., Huber, L.A., Jansen-Durr, P., 2005. Apoptosis resistance of senescent human fibroblasts is correlated with the absence of nuclear IGFBP-3. Aging Cell 4, 325–330.

[4] H.A. Cheema Nashwa, M. Debbie, M. Aiken Judd, Apoptosis and necrosis mediate skeletal muscle fiber loss in age-induced mitochondrial enzymatic abnormalities, Aging Cell 14 (6) (2015) 1085–1093.

[5] Kou, X.; Li, J.; Liu, X.; Yang, X.; Fan, J.; Chen, N. Ampelopsin attenuates the atrophy of skeletal muscle from D-gal-induced aging rats through activating AMPK/SIRT1/PGC-1α signaling cascade. Biomedicine & Pharmacotherapy 2017, 90, 311-320.

[6] Chen, Q.N.; Fan, Z.; Lyu, A.K.; Wu, J.; Guo, A.; Yang, Y.F.; Chen, J.L.; Xiao, Q. Effect of sarcolipin-mediated cell transdifferentiation in sarcopenia-associated skeletal muscle fibrosis. Exp Cell Res 2020, 389, 111890.

[7] Whitman, S.A., Wacker, M.J., Richmond, S.R., Godard, M.P., 2005. Contributions of the ubiquitin-proteasome pathway and apoptosis to human skeletal muscle wasting with age. Pflugers Arch. 450, 437–446.

  1. The major problem in my opinion is that the followed protocol used to induce senescence in C2C12 myoblasts after D-gal treatment is not properly described. Do the authors only treat cells with D-gal once for 48 h and then assess for the markers of senescence? That would mean that the observed phenomena could be rather an immediate and probably transient response of the cells to the treatment than an established senescent phenotype. In addition, when nobiletin is added? Are the cells pre-treated with nobiletin before being treated with D-gal? Are nobiletin and D-gal applied simultaneously? If D-gal treatment is short-term, this could give an explanation for the extremely low percentages of SA-beta Gal- and p16-positive cells presented in Fig. 2B and 2C. This should be checked by the authors, since a 0.8% percentage of SA-beta Gal-positive cells and a 1% percentage of p16-positive cells cannot a support the manifestation of senescence and the existence of an aging model.

--Response: Thank you for carefully reading our manuscript and giving us constructive comments, we quite agree to your comments and suggestions about the manuscript. Meanwhile, we must apologize for giving an imperfect the method of cell culture and treatment. We had revised 4.2. Cell culture and treatment, and revised text was as follows:

The C2C12 myoblasts aging model was established via D-gal (20 mg/ml) treatment for 48 h. Nob (10 μM) were dissolved in DMSO and added to the medium together with D-gal. When the cells had grown to 50%–60% confluency, they were treated as follows: (1) Control (CK) group; (2) D-gal group; (3) D-gal + Nob group.

To obtain differentiated myotubes, C2C12 myoblasts were incubated in DMEM medium with 10% FBS and 1% P/S until 80%–90% confluent. Then, C2C12 myoblasts were maintained in differentiation medium (DM) for differentiation, and then treated as follows: (1) CK group; (2) D-gal group; (3) D-gal + Nob group. Nob (10 μM) were dissolved in DMSO and added to the DM together with D-gal (20mg/ml). DM was changed every two days until the cells were fully differentiated (about day 5).

For the review's other questions, we mainly give the following three responses:

(1)In our study, Nob (10 μM) were dissolved in DMSO and added to the medium together with D-gal according to the method of Li et al [1].

[1]. Li, H.M.; Liu, X.; Meng, Z.Y.; Wang, L.; Zhao, L.M.; Chen, H.; Wang, Z.X.; Cui, H.; Tang, X.Q.; Li, X.H.; et al. Kanglexin delays heart aging by promoting mitophagy. Acta Pharmacol Sin 2022, 43, 613-623.

(2)D-gal-induced aging phenomena was long-term. Because the prominent characteristics of senescent cells are irreversible and durable growth arrest. In our study, the C2C12 myoblasts aging model was established via D-galactose-induced oxidative stress, which led to cellular damage and up-regulated the expression of P53 and P21 to enforce cell cycle arrest [2].

[2]. Maharajan, N.; Cho, G.W. Camphorquinone Promotes the Antisenescence Effect via Activating AMPK/SIRT1 in Stem Cells and D-Galactose-Induced Aging Mice. Antioxidants-Basel 2021, 10.

(3)We must apologize for my carelessness and giving wrong figures. I'm sorry that I forgot to convert the percentage when drawing because of my carelessness. I had revised Figure 2B and Figure 2D, the revised figures are shown in the following figures:

We really do not understand if it is right for our understanding about your comments, we feel very sorry for this again. We will continue to make further revisions if needed.

Round 2

Reviewer 2 Report

The authors adequately answered to the comments. It can be acceptable for publication. 

Author Response

Dear editor and the reviewer,

We sincerely thank you for taking your valuable time to revise the paper for us and giving us a golden opportunity to revise our manuscript. We appreciate the editor and reviewer very much for the positive comments and suggestions on our manuscript entitled “Nobiletin prevents D-galactose-induced C2C12 cell aging by improving mitochondrial function” (ijms-1917351).

We hope that our revised manuscript is now suitable for publication in IJMS. Looking forward to hearing from you.

Thank you for the consideration.

Sincerely,

Fazheng Ren

Reviewer 4 Report

This is the revised version of a previously submitted manuscript. The authors have addressed most of my concerns raised during the previous round of the reviewing process.

1. There are still some grammar and syntax errors that need to be corrected (for examples, please refer to the uploaded file). 

2. There are still some references about the implication of apoptosis in the induction of senescence. Non-viable cells could not become senescent. However, there are biochemical pathways, molecules and mechnaisms that can be shared by both phenomena and lead either to apoptosis or senescence.

3. I see in the .doc file that the quality of images is better. However, please consider the quality of Figures in the pdf file.

4. The authors should refer to the statistical test that followed ANOVA in order to estimate differences among experimental groups.

Author Response

Revision items sheet (No. ijms-1917351)

Point by point response for the reviewer and editor

Dear editor and the reviewer,

Thank you very much for the positive evaluation of our manuscript entitled “Nobiletin prevents D-galactose-induced C2C12 cell aging by improving mitochondrial function” (ijms-1917351). We appreciate the constructive and helpful comments of reviewer and editor; they helped to greatly improve our manuscript. To address all points raised by the reviewer and editor, we added new content in revised manuscript. The texts of our responses are marked in blue font in this file. Revised text was also shown in blue font in the revised manuscript file, and the exact line numbers where changes were pointed out in the responses.

We hope that our revised manuscript is now suitable for publication in IJMS.

Thank you for the consideration.

Sincerely,

Fazheng Ren

The responses to reviewer’s comments and the main corrections in the paper are as follows:

  1. There are still some grammar and syntax errors that need to be corrected (for examples, please refer to the uploaded file).

--Response: We do thank you for your kindly suggestion. Meanwhile, we must apologize for giving an imperfect and incomplete manuscript. We had revised some grammar and syntax errors you pointed out. In addition, we also had checked the grammar and logic of the full text, and revised text was shown in blue font in the revised manuscript file.

In line 41: We had revised “accelerates” into “accelerate”.

In line 42: We had revised “promotes” into “promote”.

In line 42: We had revised “accelerates” into “accelerate”.

In line 44-49: We had revised the sentences into “Nobiletin (Nob) is a polymethoxyl flavonoid found in some citrus peels and has been reported to have multiple functions [15], including antioxidant [16], and anti-inflammatory functions [15,17], to improve mitochondrial function, and to maintain metabolic homeostasis [3]. A previous finding study has reported that Nob enhanced antioxidant activity to clear ROS and MDA production and alleviate nonalcoholic fatty liver disease in rats fed with a high fat diet [16].”

In line 79-81: We had revised the sentences into “However, compared with D-gal group, Nob treatment significantly restored the ratio of p-S473-Akt/Akt and p-p70 S6K/p70 S6K, which increased by 56.2% and 22.6%, respectively (Figure 1C)”.

In line 89-91: We had revised the sentences into “Taken together, Nob improved D-gal-induced muscle fiber atrophy by balancing protein synthesis and with protein degradation”.

In line 101-102, 125-126, 145-146, 177-178 and 202-203: We had revised the sentences into “Different lowercase letters represent significant differences between among different treatment groups (P<0.05)”.

In line 149: We had revised “factors” into “factor”.

In line 153: We had revised “contributes” into “contribute”.

In line 160: We had deleted the “was”.

In line 241: We had deleted the “intervention”.

In line 246: We had revised the “Mitochondria” into “mitochondrial”.

In line 263: We had revised “intervention” into “treatment”.

In line 273: We had deleted the “to”.

In line 291 and 298: We had revised the “were” into “was”.

In line 374: We had revised the “prevent” into “prevented”.

  1. There are still some references about the implication of apoptosis in the induction of senescence. Non

viable cells could not become senescent. However, there are biochemical pathways, molecules and mechnaisms that can be shared by both phenomena and lead either to apoptosis or sensecence.

--Response: Thank you for carefully reading our manuscript and giving us constructive comment, we quite agree to your comment and suggestion about the manuscript. Yes, apoptosis does not induce cellular senescence. Apoptosis leads to cell death. Therefore, we had deleted the “and apoptosis” in line 38.

  1. I see in the .doc file that the quality of images is better. However, please consider the quality of Figures in the pdf file.

--Response: We do appreciate you for giving us helpful suggestion. We had revised all the figures to high resolution figures (600dpi). The revised pictures had been replaced in the revised manuscript file and uploaded in attachments.

  1. The authors should refer to the statistical test that followed ANOVA in order to estimate differences among experimental groups.

--Response: Thank you for carefully reading our manuscript and giving us constructive and helpful comment and suggestion. We quite agree to your comment and suggestion about the manuscript. We had revised the sentence into “All data were analyzed using one-way ANOVA followed by Duncan’ post-hoc test.”

We really do not understand if it is right for our understanding about your comments, we feel very sorry for this again. We will continue to make further revisions if needed.
